# Electrosynthesis of polymer-grade ethylene via acetylene semihydrogenation over undercoordinated Cu nanodots

Weiqing Xue[1,2,9], Xinyan Liu[3,9], Chunxiao Liu[2], Xinyan Zhang[1,2], Jiawei Li [1,2], Zhengwu Yang[1], Peixin Cui [4], Hong-Jie Peng [3,5], Qiu Jiang [2], Hongliang Li [1], Pengping Xu[1,6], Tingting Zheng [2] ✉, Chuan Xia [2,5,7] ✉ & Jie Zeng [1,8] ✉

The removal of acetylene impurities remains important yet challenging to the ethylene downstream industry. Current thermocatalytic semihydrogenation processes require high temperature and excess hydrogen to guarantee complete acetylene conversion. For this reason, renewable electricity-based electrocatalytic semihydrogenation of acetylene over Cu-based catalysts is an attractive route compared to the energy-intensive thermocatalytic processes. However, active Cu electrocatalysts still face competition from side reactions and often require high overpotentials. Here, we present an undercoordinated Cu nanodots catalyst with an onset potential of −0.15 V versus reversible hydrogen electrode that can exclusively convert $C_2H_2$ to $C_2H_4$ with a maximum Faradaic efficiency of ~95.9% and high intrinsic activity in excess of −450 mA cm$^{-2}$ under pure $C_2H_2$ flow. Subsequently, we successfully demonstrate simulated crude ethylene purification, continuously producing polymer-grade $C_2H_4$ with <1 ppm $C_2H_2$ for 130 h at a space velocity of $1.35 \times 10^5$ ml g$_{cat}^{-1}$ h$^{-1}$. Theoretical calculations and in situ spectroscopies reveal a lower energy barrier for acetylene semihydrogenation over undercoordinated Cu sites than nondefective Cu surface, resulting in the excellent $C_2H_2$-to-$C_2H_4$ catalytic activity of Cu nanodots.

Ethylene ($C_2H_4$) is the primary building block in the production of commercially useful commodity chemicals like plastics, antifreeze, fibers, organic solvents, etc.[1–3]. Polyethylene plastic, produced by the *Ziegler-Natta* polymerization process, is one of the most important downstream products of $C_2H_4$. It is worth noting that such a process requires polymer-grade $C_2H_4$ as the feedstock because even 0.5% acetylene ($C_2H_2$) impurities in raw ethylene streams can irreversibly poison the *Ziegler-Natta* catalysts, resulting in much nerfed catalytic

[1]Hefei National Research Center for Physical Sciences at the Microscale, Key Laboratory of Strongly-Coupled Quantum Matter Physics of Chinese Academy of Sciences, Key Laboratory of Surface and Interface Chemistry and Energy Catalysis of Anhui Higher Education Institutes, Department of Chemical Physics, University of Science and Technology of China, 230026 Hefei, Anhui, P. R. China. [2]School of Materials and Energy, University of Electronic Science and Technology of China, 611731 Chengdu, P. R. China. [3]Institute of Fundamental and Frontier Sciences, University of Electronic Science and Technology of China, 611731 Chengdu, P. R. China. [4]Key Laboratory of Soil Environment and Pollution Remediation, Institute of Soil Science, Chinese Academy of Sciences, 210008 Nanjing, P. R. China. [5]Yangtze Delta Region Institute (Huzhou), University of Electronic Science and Technology of China, 313001 Huzhou, Zhejiang, P. R. China. [6]Institute of Advanced Technology, University of Science and Technology of China, 230031 Hefei, Anhui, P. R. China. [7]Research Center for Carbon-Neutral Environmental & Energy Technology, University of Electronic Science and Technology of China, 611731 Chengdu, P. R. China. [8]School of Chemistry & Chemical Engineering, Anhui University of Technology, 243002 Ma'anshan, Anhui, P. R. China. [9]These authors contributed equally: Weiqing Xue, Xinyan Liu. ✉e-mail: ttzheng@uestc.edu.cn; chuan.xia@uestc.edu.cn; zengj@ustc.edu.cn

activity[4]. Accordingly, to perform this reaction economically at scale in the long term, the $C_2H_2$ impurities in the $C_2H_4$ stream must be lower than 5 parts per million (ppm) to satisfy the polymer-grade $C_2H_4$ requirement[5]. Despite the widespread use of polymer-grade $C_2H_4$ as industrial raw materials, its commercial synthesis still faces inefficient and energy-intensive purification steps.

Typically, solvent absorption and thermally catalyzed semihydrogenation of $C_2H_2$ are the two main strategies to remove $C_2H_2$ impurities from ethylene streams. Solvent absorption is the purification approach in the early years, in which $C_2H_2$ is extracted by solvents such as *N,N*-dimethylformamide, *N*-methylpyrrolidinone, or ethyl acetate. Unfortunately, such a method is not environmentally sustainable due to high-cost organic solvent consumption and $C_2H_4$ loss[6,7]. Even though emerging porous sorbents, especially metal-organic frameworks, exhibit the potential for $C_2H_2/C_2H_4$ separation[8–12], it also suffers from several drawbacks, such as poor stability and the trade-off between adsorption capacity and selectivity[1]. Currently, thermocatalytic semihydrogenation of $C_2H_2$ over Pd-based catalysts is widely used for the industrial-scale synthesis of polymer-grade $C_2H_4$[13–20], but it requires relatively high operating temperatures (100–250 °C). In addition, excess hydrogen ($H_2$) is needed to guarantee this conversion process, thereby bringing safety problems and leading to over-hydrogenation. Further, an energy-intensive downstream separation is often required to obtain clean $C_2H_4$. Therefore, a cost-effective and sustainable strategy is desirable to purify the crude $C_2H_4$ into polymer-grade feedstocks.

Electrocatalytic acetylene semihydrogenation (EASH) is an appealing alternative approach ($C_2H_2 + 2H_2O + 2e^- \rightarrow C_2H_4 + 2OH^-$) powered by renewable electricity under ambient conditions. EASH can be carried out in aqueous media, and water ($H_2O$) serves as the proton source rather than excess $H_2$, which is in line with the concept of safety and sustainability. Nonetheless, due to the poor solubility of acetylene and lack of efficient catalysts, EASH has been stagnant since its appearance in the 1970s[21–23]. Recent works have addressed the solubility issue using gas diffusion layer (GDL) electrodes, where the mass transfer limitations can be broken by constructing a triple-phase boundary[24–27]. However, it is still essential to deal with the competing hydrogen evolution reaction (HER), over-hydrogenation, and C–C coupling. Copper (Cu) has been recognized as one of the predominant catalysts for activating acetylene while selectively producing $C_2H_4$. Despite the vigorous efforts in engineering Cu catalysts towards efficient EASH, these reported Cu catalysts required high overpotentials[24–26], translating to lower energy efficiency, and failed to completely remove the $C_2H_2$ impurities (<1 ppm) from the crude ethylene flow.

Here, we designed undercoordinated Cu nanodots (Cu NDs) as highly efficient EASH catalyst, which was derived from in situ reduction supplied with $C_2H_2$ gas. Under pure acetylene flow in a three-electrode flow cell, the Cu NDs exhibited a relatively positive onset potential of −0.15 V vs. RHE and achieved a maximum ethylene Faradaic efficiency ($FE_{C2H4}$) of 95.9% at −0.69 V vs. RHE. The $FE_{C2H4}$ remained above 90.4% with a $C_2H_4$ partial current density of −452 mA cm$^{-2}$, superior to the reported catalysts[3,24–26]. Theoretical calculations and in situ spectroscopies revealed that the impressive catalytic activity of Cu NDs could be ascribed to its lower rate-determining-step energy barrier for acetylene semihydrogenation on undercoordinated Cu sites. We further showcased the continuous production of polymer-grade $C_2H_4$ from simulated crude gas that contains 0.5% acetylene. Under a flow rate of 50 standard cubic centimeter per minute (sccm), corresponding to a space velocity of $1.35 \times 10^5$ ml $g_{cat}^{-1}$ h$^{-1}$, our homemade reactor with 25 cm$^2$ electrode area could continuously generate ultrapure $C_2H_4$ ($C_2H_2$ < 1 ppm) for at least 130 h.

## Results

### Structural characterization of the Cu NDs electrocatalyst

We first synthesized the copper precursor using a facile hydrolysis method (see the "Methods" section), which yielded carbon-supported $Cu_2Cl(OH)_3$ nanoparticles (Supplementary Figs. 1–5). To form the Cu NDs catalyst, we sprayed the precursor onto the GDL, which was subjected to the in situ reduction under a constant current density of −100 mA cm$^{-2}$ for 30 min in a standard three-electrode flow cell under pure $C_2H_2$ flow. The as-reduced samples were then peeled off from the GDL for further structural characterization. Transmission electron microscopy (TEM) and high-angle annular dark-field scanning transmission electron microscopy (HAADF-STEM) images, in conjunction with energy dispersive X-ray (EDX) elemental mapping, clearly showed the uniform dispersion of Cu nanodots on activated carbon particles, with an average size of 4.4 ± 0.6 nm (Fig. 1a, b). As shown in the high-resolution TEM (HRTEM) image (Fig. 1c), the lattice fringe spacing of 0.208 nm marked on the nanodots can be indexed to the (111) plane of Cu[28]. The X-ray diffraction (XRD) and selected area electron diffraction (SAED) (Supplementary Figs. 6, 7) further revealed a pure metallic Cu crystal structure for the as-reduced Cu NDs[29]. X-ray photoelectron spectroscopy (XPS) reconfirmed the Cu (0) valence state of the Cu NDs (Fig. 1d and Supplementary Fig. 8)[30]. The mass loading of Cu in the carbon-supported Cu NDs catalyst was determined to be 29.04 wt% by inductively coupled plasma atomic emission spectroscopy (ICP-AES).

In an effort to investigate the electronic structure and coordination environment of Cu NDs under EASH conditions, we then conducted in situ X-ray absorption spectroscopy (XAS) measurements. Figure 1e shows the Cu *K*-edge normalized X-ray absorption near-edge spectroscopy (XANES) profiles of Cu NDs at different working potentials, along with Cu foil, $Cu_2O$, and CuO as references. The white line of Cu NDs at different applied potentials resembled that of copper foil, suggesting that Cu in NDs catalyst maintained a metallic feature under EASH condition[31]. The results of XANES deconvolution by linear combination fitting of the spectra of $Cu^0$ and $Cu^+$ also confirmed the active sites in the EASH were metallic Cu nanodots (Supplementary Fig. 9). As shown in the extended X-ray absorption fine structure (EXAFS) spectra (Fig. 1f and Supplementary Fig. 10), the prominent peaks of Cu NDs under different working potentials locate at approximately 2.21 Å, which was ascribed to the Cu−Cu bonds. In contrast, the spectrum of the precursor showed characteristic fingerprint peaks of Cu−O, Cu−Cl, and Cu−Cu coordination. These results were also corroborated by wavelet transform analysis and the fitting data (Supplementary Fig. 11 and Supplementary Table 1). It is worth noting that the Cu−Cu coordination number of Cu NDs at different applied potentials was -10, lower than that of 12 for Cu foil (Supplementary Table 1), implicating the unsaturated coordination of Cu sites for Cu NDs. The above results, taken together, suggest that the active phase of our in situ formed Cu NDs under EASH conditions is metallic Cu featuring undercoordinated sites, which could be represented by stepped Cu (211)[32].

### Acetylene electroreduction under pure acetylene flow

The acetylene-to-ethylene performance of the Cu NDs was first evaluated under pure acetylene flow at the flow rate of 30 sccm in a typical three-electrode flow cell, with 1 M KOH as the electrolyte (see the "Methods" section, Supplementary Figs. 12, and 13). As revealed by linear sweep voltammetry (LSV) in Supplementary Fig. 14, the Cu NDs afforded a notably higher current density under pure acetylene than that under Ar, indicating the participation of $C_2H_2$ in the reaction. Steady-state chronopotentiometry of acetylene electrolysis was recorded under different current densities from −100 to −500 mA cm$^{-2}$. The gas and liquid products at different current densities were analyzed and quantified by gas chromatography (GC) and nuclear magnetic resonance (NMR), respectively. We found that $C_2H_4$ was the main product over Cu NDs during EASH, along with a small amount of $C_4$ olefin and negligible $H_2$ (Fig. 2a and Supplementary Table 2). No other liquid products were detected according to the $^1$H-NMR spectrum (Supplementary Fig. 15). For Cu NDs, a high plateau of $C_2H_4$ Faradaic efficiency over 90% was retained under a broad potential range from −0.45 to

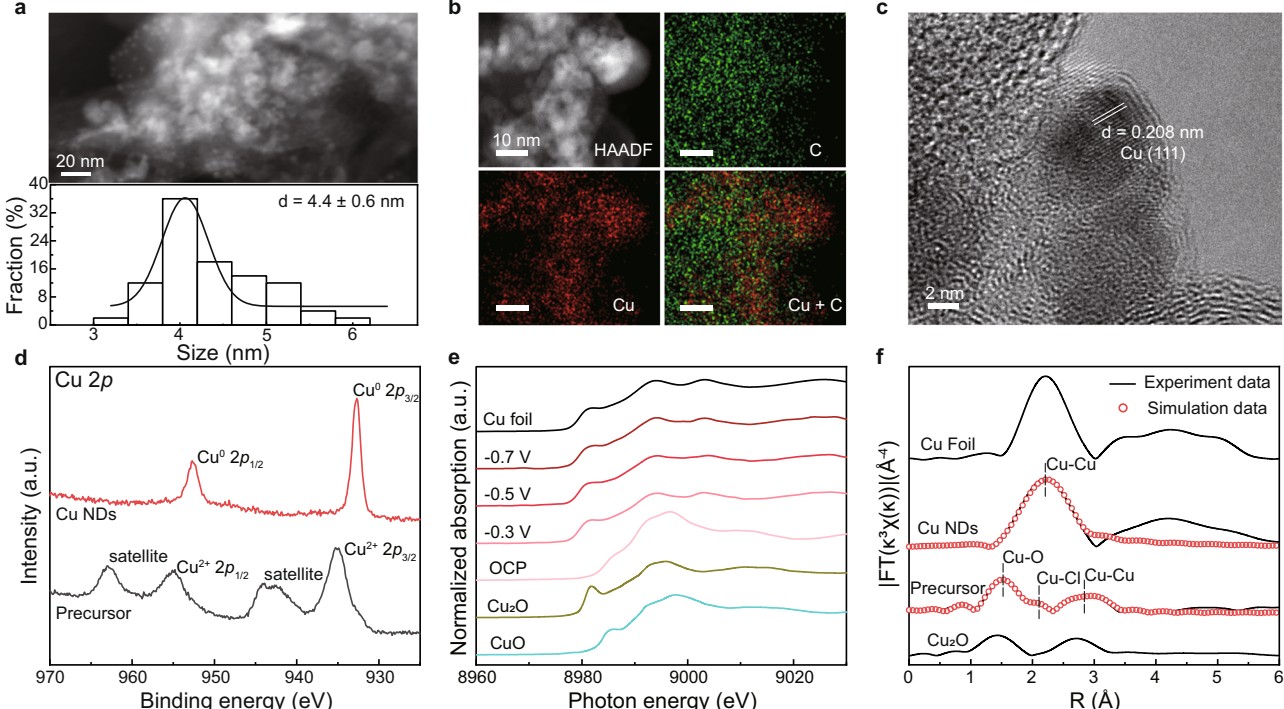

**Fig. 1 | Structural characterization of the Cu NDs catalyst. a** HAADF-STEM image of Cu NDs (up) and the corresponding size distribution histograms (down). **b** STEM-EDX mapping of the Cu NDs catalyst. **c** HRTEM image of the Cu NDs catalyst. **d** Cu 2*p* XPS spectra of the precursor and Cu NDs catalyst. **e** Normalized in situ XANES spectra of the Cu NDs catalyst under EASH conditions at different working potentials vs. RHE, along with the spectra of Cu foil, Cu₂O, and CuO as references. OCP, open-circuit potential. **f** Cu *K*-edge phase-uncorrected EXAFS spectra and the corresponding simulation curves in R space of the precursor and Cu NDs catalyst, along with the spectra of Cu foil and Cu₂O as references.

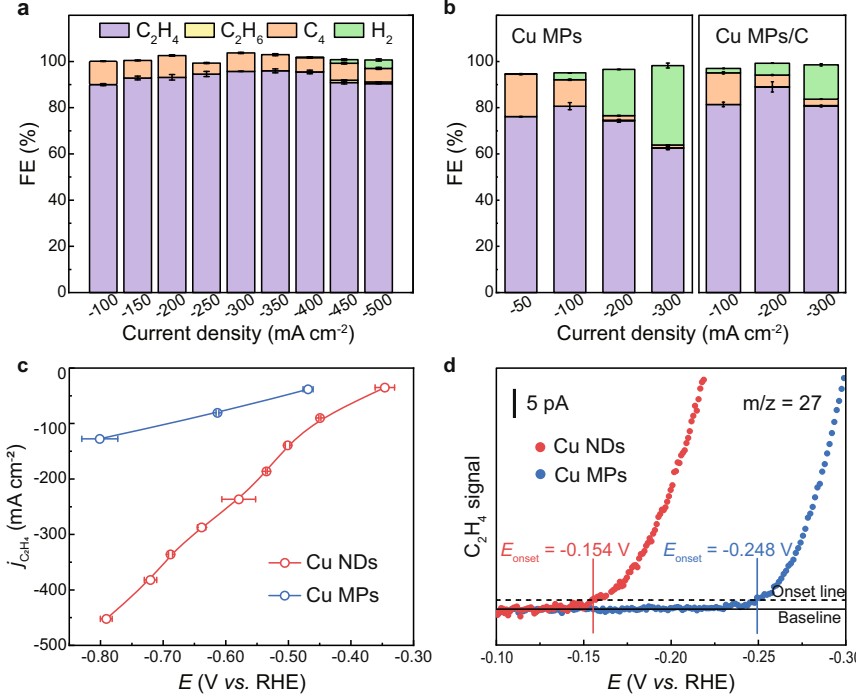

**Fig. 2 | Electrocatalytic acetylene semihydrogenation performance over Cu-based electrocatalysts under pure acetylene flow. a, b** FEs of EASH products at different current densities of **a** Cu NDs and **b** Cu MPs, Cu MPs/C. **c** Variation in the ethylene partial current density against applied potential over Cu NDs and Cu MPs. **d** In situ DEMS measurement of C₂H₄ production during EASH over Cu NDs and Cu MPs. All tests were conducted using a three-electrode flow cell in 1 M KOH solution at room temperature under pure acetylene flow. The error bars correspond to the standard deviation of at least three independent measurements.

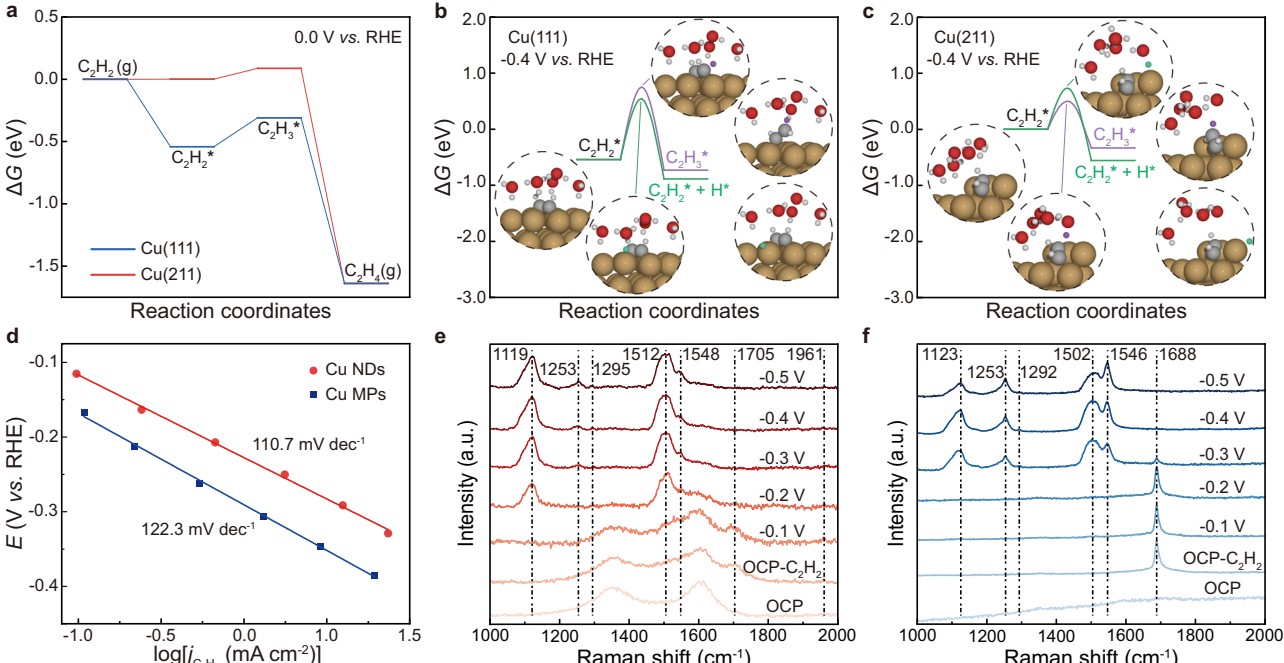

**Fig. 3 | DFT theoretical calculations and in situ electrochemical Raman study.**
**a** Free energy diagram of reducing $C_2H_2$ on Cu (111) and Cu (211) surfaces.
**b**, **c** Comparison between the barriers of forming $C_2H_3^*$ and $H^*$ on **b** (111) and
**c** Cu (211) surfaces. All energies were referenced to $C_2H_2$ and a computational
hydrogen electrode (CHE). **d** Tafel plots for EASH over Cu NDs and Cu MPs. **e**, **f** In
situ electrochemical Raman spectra of **e** Cu NDs and **f** Cu MPs in a 1 M KOH solution
at different working potentials vs. RHE without IR compensation. OCP open-circuit
potential.

−0.79 V vs. RHE. The maximum $C_2H_4$ FE of ~95.9% was achieved at a
current density of −350 mA cm⁻² under −0.69 V vs. RHE. Notably, the FE
of the competitive HER was suppressed to below 0.1% until the current
density ascended to −400 mA cm⁻². In addition, Cu NDs exhibited
excellent stability under EASH conditions (Supplementary Fig. 16) and
the effect of copper mass loading in Cu NDs on $C_2H_2$-to-$C_2H_4$ perfor-
mance was also evaluated in Supplementary Fig. 17.

For comparison, the EASH catalytic activities of Cu micro-
particles (Cu MPs, Supplementary Figs. 18, 19) and Cu MPs loaded on
activated carbon (Cu MPs/C) were investigated under the same test
conditions as those of Cu NDs (Fig. 2b and Supplementary Fig. 20). In
the case of Cu MPs, the maximum $C_2H_4$ FE was 85.0% at the current
density of −150 mA cm⁻², which decreased rapidly as the current
density increased due to HER competition, with the $H_2$ FE in excess of
34% at −300 mA cm⁻². The same trend was observed for Cu MPs/C,
suggesting activated carbon particles did not contribute to the
improvement of EASH performance. Figure 2c depicts the ethylene
partial current density ($j_{C2H4}$) at different potentials of Cu NDs and
Cu MPs. The curves showed that the catalytic activity of Cu NDs was
significantly higher than that of Cu MPs under the same working
potential, as exemplified by the higher $j_{C2H4}$ of Cu NDs. Notably, an
impressive $j_{C2H4}$ of Cu NDs was achieved as high as −452 mA cm⁻² at
−0.79 V vs. RHE, while the $C_2H_4$ FE remained above 90%. We also
conducted in situ differential electrochemical mass spectrometry
(DEMS) over Cu NDs and Cu MPs under pure acetylene, where the
onset potential of Cu NDs was about −0.15 V vs. RHE, which was ~94 mV more positive than that of Cu MPs (Fig. 2d). This comparison
presented that the undercoordinated Cu NDs catalyst exhibited
higher intrinsic activity for EASH than its bulk counterparts. The
same conclusion could be drawn from LSV curves, which also sug-
gested the negligible contribution of GDL to EASH activity (Supple-
mentary Fig. 21). As shown in Supplementary Fig. 22 and
Supplementary Table 3, our Cu NDs catalyst shows superiorities with
respect to high $j_{C2H4}$ and high $FE_{C2H4}$ compared with previously
reported catalysts[3,21,23–26,33,34].

## Study of the reaction mechanism

Density functional theory (DFT) simulations were first carried out to
elucidate the reaction mechanism of $C_2H_2$ electroreduction on Cu.
Herein, two types of surfaces were investigated: the Cu (111) surface
representing the flat, nondefective surface and the stepped Cu (211)
surface featuring undercoordinated sites created by either inherent
crystal orientations or local defects[32]. The reaction free energy of
forming $C_2H_4$ from $C_2H_2$ was first compared in Fig. 3a under 0 V vs.
RHE, equivalent to a computational hydrogen electrode (CHE) by
definition. For both surfaces, the potential limiting step (PLS) is the
reduction of adsorbed $C_2H_2^*$ to $C_2H_3^*$ because of the relatively strong
adsorption of $C_2H_2^*$ on both surfaces. In particular, Cu (111) exhibited
significantly stronger stabilization to $C_2H_2^*$ than Cu (211) and more
negative potential is required for driving the above-identified PLS. In
other words, undercoordinated Cu sites were able to catalyze $C_2H_2$
electroreduction at lower overpotentials than terrace Cu sites, which
was consistent with the DEMS results (Fig. 2d). To further understand
the $C_2H_4$ selectivity against HER, the kinetic barriers of proton transfer
to $C_2H_2^*$ and forming $C_2H_3^*$ were compared with the barriers of
transferring a proton to the surface and forming $H^*$ (Fig. 3b, c). The as-
formed $H^*$ then produced $H_2$ through either Heyrovsky or Tafel
pathways. It was clear that under −0.4 V vs. RHE, a potential where a
reasonably large current density was observed experimentally, the
barrier of forming $C_2H_3^*$ on the Cu (211) surface (0.49 eV) was con-
siderably lower than that on the Cu (111) surface (1.29 eV). This barrier
was also smaller than the barrier of forming $H^*$ on the Cu (211) surface
(0.72 eV). In contrast, an inverse trend was observed for the Cu (111)
surface. The same conclusion could also be drawn from the Tafel plots,
where the EASH over Cu NDs and Cu MPs have the same rate-
determining step (110.7 mV dec⁻¹ for Cu NDs and 122.3 mV dec⁻¹ for Cu
MPs), and the lower Tafel slope on Cu NDs indicates that the hydro-
genation of adsorbed $C_2H_2^*$ to $C_2H_3^*$ has faster reaction kinetics
(Fig. 3d). Thus, the undercoordinated sites of Cu are identified as the
active sites for selective $C_2H_2$ electroreduction against the HER. As
disclosed by the XAFS fitting results (Supplementary Table 1), Cu NDs

feature abundant undercoordinated sites, thus rendering high activity for selective $C_2H_2$ electroreduction. To rule out the effect of surface area and obtain a clearer idea of the intrinsic activity, we conducted Pb underpotential deposition (UPD) to obtain the electrochemically active surface area (ECSA) (Supplementary Fig. 23)[35]. The ECSA-normalized current densities of Cu NDs were still higher than those of Cu MPs under the same working potential, indicating the superior intrinsic catalytic activity of Cu NDs over Cu MPs (Supplementary Fig. 24). Moreover, Cu NDs intrinsically favored selective $C_2H_4$ production, while Cu MPs was more preferable to HER (Supplementary Fig. 25).

We further conducted in situ electrochemical Raman spectroscopy to monitor the EASH processes on these Cu catalysts (Fig. 3e, f). For Cu NDs, at open-circuit potential (OCP), the peaks at approximately 1350 and 1610 $cm^{-1}$ belonged to the D-Peak and G-Peak of carbon[36], respectively. Once $C_2H_2$ was introduced, a fingerprint peak corresponding to absorbed $C_2H_2$* appeared at 1705 $cm^{-1}$, which exhibited a red-shift relative to $C_2H_2$ (1961 $cm^{-1}$)[37], associated with the σ-π-$C_2H_2$ adsorbed configuration[38]. When increasing the potentials for Cu NDs, the intensity of the $C_2H_2$* peak decreased gradually, suggesting that $C_2H_2$ was being consumed (Fig. 3e). Upon applying the cathodic potential to −0.2 V vs. RHE, the π-d type adsorbate-surface ethylene signal arose, including 1548 $cm^{-1}$ for the C = C stretching mode and 1295 $cm^{-1}$ for the $CH_2$ bending mode[39]. Beyond that, additional peaks were observed at 1119, 1253, and 1512 $cm^{-1}$, which were assigned to the C–C, C–H, and C = C vibrations of polyacetylene[38], respectively (Fig. 3e). In contrast, the signal of $C_2H_2$* over Cu MPs was observed at 1688 $cm^{-1}$ (Fig. 3f), indicating the same adsorbed configuration but a stronger adsorption strength than Cu NDs, conforming to the DFT simulation results (Fig. 3a). In addition, the ethylene vibration signals over Cu MPs were observed at 1546 and 1292 $cm^{-1}$, which were red-shifted compared with that over Cu NDs, suggesting the stronger adsorption of $C_2H_3$* over Cu MPs. These results indicated that ethylene was more readily desorbed from Cu NDs than Cu MPs and thus Cu NDs had better ethylene selectivity.

### Electrosynthesis of polymer-grade $C_2H_4$

Considering that the concentration of acetylene impurities in industrial crude ethylene is generally between 0.5% and 3%, we, therefore, employed an ethylene-rich gas source (0.5% $C_2H_2$, 20% $C_2H_4$ balanced with Ar) to simulate the crude ethylene. A three-electrode flow cell was first used to evaluate the EASH catalytic activity over different catalysts at low acetylene partial pressure. As shown in Supplementary Fig. 26, at the flow rate of 20 sccm, a peak $C_2H_2$ conversion of 67.2% was afforded over Cu NDs, which was higher than that of 43.4% over Cu MPs. Ethylene was the major product of the EASH reaction for Cu NDs, with the highest $C_2H_4$ FE of 92.0% at −10.0 mA, while HER was suppressed over Cu NDs to a greater extent than over Cu MPs.

Taking advantage of low resistance, low energy consumption, and compact configuration, membrane electrode assembly (MEA) reactors are promising for performing practical gas-phase reactions. As such, we designed an MEA-type two-electrode reactor (Fig. 4a) for the continuous generation of polymer-grade $C_2H_4$. Figure 4b shows the EASH products distribution over Cu NDs at different currents, along with the corresponding $C_2H_2$ conversion rate, in a two-electrode MEA reactor (electrode area, 4 $cm^2$) at a flow rate of 10 sccm. When a current of −7.50 mA was applied (theoretical limiting conversion current is −6.98 mA, see the "Methods" section), acetylene could be completely reduced (<1 ppm, Supplementary Fig. 27) with 92.6% $C_2H_4$ selectivity, while HER was entirely stifled (Supplementary Fig. 28). Remarkably, our two-electrode MEA reactor can steadily operate for as long as 70 h at −7.50 mA without noticeable performance decay, continuously providing ultrapure $C_2H_4$ with negligible gas impurities (<1 ppm for acetylene, ethane, and hydrogen, Fig. 4c). In addition, Cu NDs maintained its

morphology, phase, and valence state after the stability test, indicating its excellent material stability (Supplementary Fig. 29).

To validate the scalability of our MEA device and achieve a higher ethylene purification efficiency, we extended the geometric electrode area from 4 $cm^2$ used for performance evaluation to 25 $cm^2$ in one-unit modular cell (Supplementary Fig. 30). As depicted in Fig. 4d, the enlarged MEA reactor could completely convert $C_2H_2$ impurities to $C_2H_4$ at different flow rates from 10 to 50 sccm to meet various production requirements. In an aim to remove the acetylene as thoroughly as possible, we applied a current slightly higher than the corresponding theoretical limiting conversion current at different flow rates. This showed that with the increase in flow rate, the selectivity of ethylene took on an upward trend, where the ethylene selectivity remained above 90% at a flow rate >20 sccm and reached 93.6% at 50 sccm, corresponding to a space velocity of $1.35 \times 10^5$ ml $g_{cat}^{-1}$ $h^{-1}$. In addition, because of the relatively positive onset potential, Cu NDs showed a cell voltage of −1.89 V at 50 sccm, leading to improved reactor energy efficiency. Higher flow rates also resulted in a slight increase of hydrogen volume in the ethylene flow, presumably due to the shortened gas residence time and the applied higher current, which both encouraged the HER (Supplementary Table 4). Notably, our enlarged two-electrode reactor can continuously operate for 130 h at −50.0 mA with negligible performance decay (residual $C_2H_2$ < 1 ppm, $H_2$ volume ≈ 0.18%, cell voltage maintained at approximately −1.89 V), producing ultrapure $C_2H_4$ along with 1800 ppm $H_2$ and 370 ppm $C_4$ species (Fig. 4e), which can be easily separated using the current cryogenic liquefaction apparatus (Supplementary Fig. 31). As shown in Fig. 4f, Supplementary Fig. 32, Supplementary Tables 5, and 6, Cu NDs exhibits distinct advantages over reported state-of-the-art catalysts in $C_2H_2$ removal. These results cast light on the potential of electrocatalytic acetylene semihydrogenation over Cu NDs for the production of a polymer-grade ethylene feed. Moreover, the modularity and ease of scale-up of the MEA reactor used in our work provide an unprecedented opportunity to accelerate the industrialization of this approach (Supplementary Note 1).

## Discussion

By virtue of the experimental and theoretical results, we demonstrated that the undercoordinated Cu NDs catalyst exhibited remarkable catalytic activity for EASH with a relatively positive onset potential of −0.15 V vs. RHE and a high $FE_{C2H4}$ of 90.4% under the high $C_2H_4$ partial current density of −452 mA $cm^{-2}$. Moreover, we showcased the electrocatalytic purification of crude ethylene via a homemade two-electrode MEA reactor at a space velocity of $1.35 \times 10^5$ ml $g_{cat}^{-1}$ $h^{-1}$ and resulted in negligible acetylene impurities (<1 ppm). Our work implicates a great potential for replacing the current energy-intensive thermal catalysis process and provides a sustainable avenue for highly efficient electricity-powered acetylene-to-ethylene conversion.

## Methods

### Chemicals

All chemicals were used without further purification. Copper (II) chloride ($CuCl_2$), copper sulfate pentahydrate ($CuSO_4·5H_2O$), ascorbic acid (AA), sodium hydroxide (NaOH), potassium hydroxide (KOH), sodium chloride (NaCl), ethanol (>99.7%), isopropanol (>99.7%), diethanolamine (DEA > 99%), ethylene glycol (EG > 99%) were purchased from Macklin. Polyvinylpyrrolidone (PVP, K29-K32) and Nafion (5 wt%) were purchased from Sigma-Aldrich. Activated carbon (Vulcan XC-72) was obtained from SCI Materials Hub. Deionized (DI) water was used throughout the experiments.

### Preparation of Cu NDs

For the synthesis of carbon-supported $Cu_2Cl(OH)_3$ precursor, 400 mg of activated carbon (XC-72) was dispersed in 80 ml of ethanol in a 250 ml flask with sonication for 30 min. After that, 1 g of $CuCl_2$ was added into the flask and stirred for another 30 min to get a uniform ink.

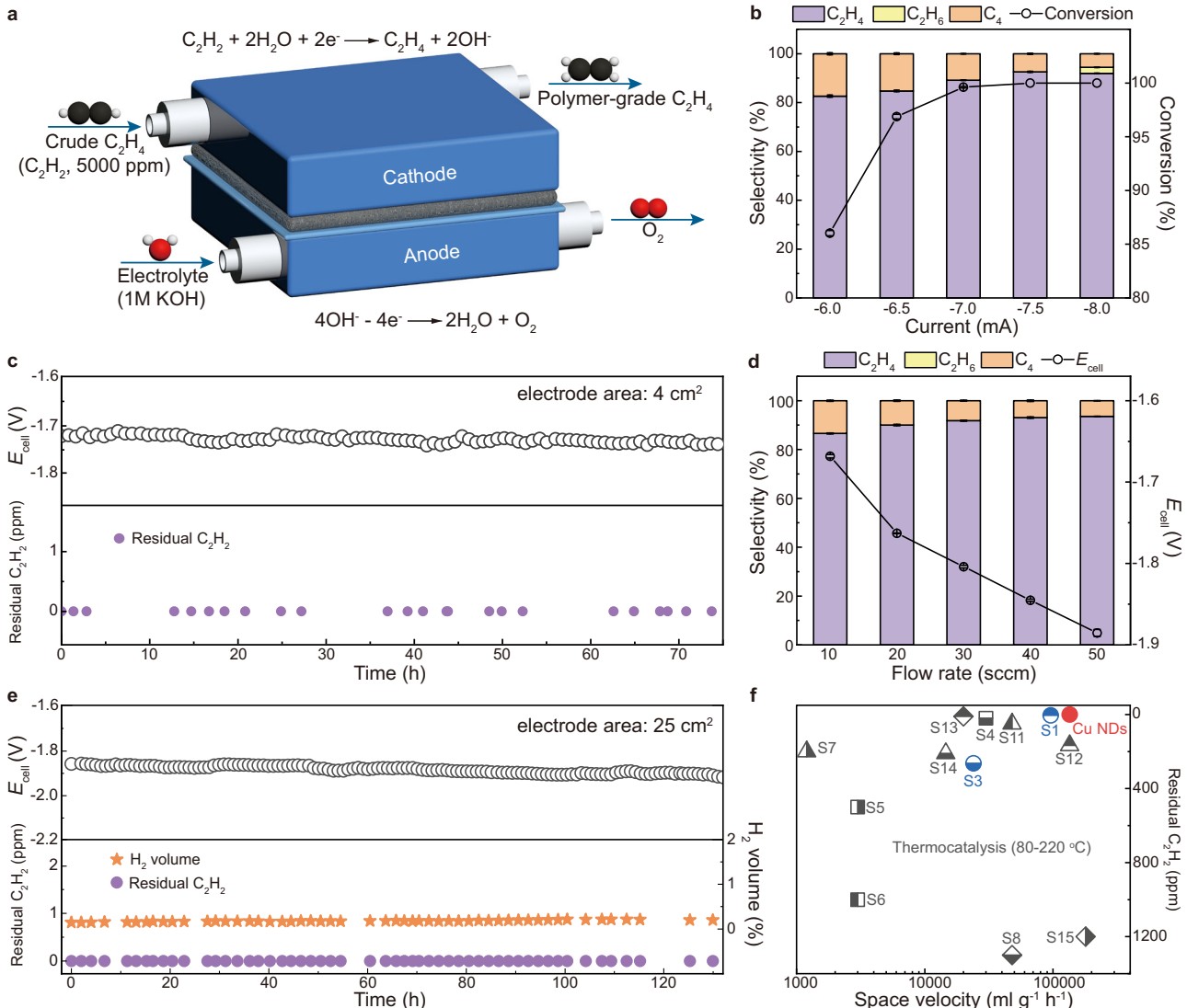

**Fig. 4 | Acetylene removal in ethylene-rich simulated gas (0.5% $C_2H_2$, 20% $C_2H_4$ balanced with Ar).** **a** Schematic of the homemade MEA-type two-electrode reactor. Between the cathode and anode were the GDL loaded with catalyst and anion exchange membrane (AEM). **b** Selectivity of EASH products at different currents and corresponding $C_2H_2$ conversion over Cu NDs in the two-electrode reactor (electrode area, 4 cm²) at a flow rate of 10 sccm. **c** Long-term operation of EASH at a constant current of −7.50 mA under the reaction conditions of (**b**). **d** Selectivity of EASH products at different flow rates and the corresponding full-cell potential over

Cu NDs in the homemade 25 cm² two-electrode reactor. Applied current: 10 sccm, −8.0 mA; 20 sccm, −18.0 mA; 30 sccm, −28.0 mA; 40 sccm, −39.0 mA; 50 sccm, −50.0 mA. **e** Durability test for 130 h conducted in the homemade 25 cm² two-electrode reactor at the current of −50 mA with the flow rate of 50 sccm. **f** Comparison of the acetylene semihydrogenation performance over Cu NDs with previously reported state-of-the-art catalysts. The balls represent the electro-catalytic processes while the others were the thermocatalysis. All comparison data are from the references summarized in Supplementary Table 5.

---

Diethanolamine (DEA) solution (1 g of DEA dissolved in 5 ml of ethanol) was then slowly dropped into the flask for 10 min and refluxed at 100 °C for 1 h, under stirring. After cooling to room temperature, 100 ml of DI water was added into the flask and sonicated for 30 min to completely hydrolyze the copper complex. The precursor was collected by centrifugation at 15,800 × *g* for 10 min and washed three times with DI water and two times with ethanol, respectively, and then dried at 70 °C for 24 h. Cu NDs catalyst was then in situ formed by reducing the precursors at a constant current density of −100 mA cm⁻² for 30 min in a standard three-electrode flow cell system supplied with $C_2H_2$ gas at a flow rate of 30 sccm, using 1 M KOH as electrolyte. The mass loading of copper in the Cu NDs can be controlled easily by adjusting the amount of $CuCl_2$ in the feed.

### Preparation of Cu MPs
To synthesize the Cu MPs, 125 mg of $CuSO_4 \cdot 5H_2O$ and 200 mg of PVP were dissolved in a 50 ml flask containing 20 mL of EG. Then added

0.2 mL of NaCl EG solution (1 M) into the flask and heated to 100 °C. Next, another mixture of 1 mL of NaOH EG solution (1 M) and 2.5 mL of AA EG solution (0.5 M) was slowly injected into the flask. Then the mixture was heated for 40 min at 100 °C. After that, Cu MPs were cleaned and collected using a mixture of ethanol and water by centrifugation.

### Preparation of working electrodes
30 mg of the catalyst precursor powder was dispersed in 6 ml of iso-propanol with sonication for 1 h. Then, 40 µl of Nafion solution (5 wt%) was added into the solution for another 30 min. After that, the precursor ink was air-brushed onto a piece of 4 × 4 cm² GDL (YLS-30T) to obtain the working electrode. The mass loading of catalysts precursors on GDL was controlled to be 1 mg cm⁻², after the effect of mass loading per geometric area on acetylene-to-ethylene performance was evaluated by controlling the mass loading of precursors on GDL at 0.49, 0.89, and 2.02 mg cm⁻², respectively (Supplementary Figs. 12 and 13).

## Electrochemical measurements

All electrochemical measurements were conducted at room temperature. CHI (1140c) and Bio-Logic (VSP-3e) electrochemical workstations were employed for the electrochemical measurements. The Ag/AgCl (saturated KCl) was adopted as the reference electrode, and the counter electrode was Ni foam for the oxygen evolution reaction. All potentials were converted to the RHE reference scale using the relation below and the solution resistance was compensated with an 80% compensation coefficient unless otherwise mentioned.

$$E_{RHE} = E_{Ag/AgCl} + 0.198 + 0.059 \times pH \tag{1}$$

For a three-electrode flow cell system under pure $C_2H_2$, a customized reactor with an electrode area of 0.75 cm$^2$ was designed. The cathode gas chamber was supplied with pure $C_2H_2$ at a flow rate of 30 sccm, when 1 M KOH was pumped around both the cathode and anode side at a flow rate of 4 ml min$^{-1}$. A Nafion 115 membrane (Fuel Cell Store) was sandwiched between an anolyte and electrolyte to separate the chambers.

For three-electrode flow cell systems under ethylene-rich simulated gas (0.5% $C_2H_2$, 20% $C_2H_4$ balanced with Ar), the experimental conditions were consistent with the above except that the flow rate was 20 sccm.

For two-electrode reactors, two types of electrode area (2 × 2 and 5 × 5 cm$^2$) were used to purify ethylene. Feed gas (0.5% $C_2H_2$, 20% $C_2H_4$ balanced with Ar) was delivered to the GDL from the back side of the cathode at the different flow rate, while 1 M KOH aqueous solution was circulated around the anode side at a flow rate of 10 ml min$^{-1}$. An anion-exchange membrane (AEM, Dioxide Materials and Membranes International) was used for ion exchange.

In situ DEMS was conducted using a homemade flow cell and the mass spectrometer was equipped with a capillary injection port. The scan rate of LSV measurement was set to 5 mV s$^{-1}$. The signal of mass-to-charge ratio of 27 and 26 was ascribed to $C_2H_4$ and $C_2H_2$, respectively. The onset line was determined according to the positions where the signal-to-noise ratio reached 5.

In situ Raman analyses were performed using the Renishaw inVia Raman analyzer equipped with 532 nm laser, combined with the custom flow cell. While catalysts loaded GDL served as the working electrodes, Pt wire and Ag/AgCl (saturated KCl) were used as counter and reference electrodes, respectively. During experiments, the laser was focused on the surface of the sample with a laser intensity of 5 mW.

Pd UPD was conducted in an Ar-saturated solution of 100 mM $HClO_4$ + 1 mM $Pb(ClO_4)_2$, using Pt wire as the anode electrode. The cathode was held at −0.12 V vs. RHE for 1 min and then LSV was taken from −0.12 to 0.10 V vs. RHE at a scan rate of 5 mV s$^{-1}$. The background was recorded in 100 mM $HClO_4$ without $Pb(ClO_4)_2$. The ECSA was calculated from the total charge of Pb monolayer stripping from Cu surface with a conversion factor of 310 μC cm$^{-2}$.

## Data analysis

The gas products ($H_2$, $C_2H_2$, $C_2H_4$, $C_2H_6$, $C_4$) were analyzed using a gas chromatograph (GC, Agilent 8990) coupled with a thermal conductivity detector and flame ionization detector. The calibration curve across the zero point confirmed that the detection concentration limit of GC was as low as 1 ppm (Supplementary Fig. 27). The FE of the gas products was calculated through the concentration ($x$) detected by GC according to the equation:

$$FE(\%) = \frac{nFx\upsilon}{V_mI} \times \frac{1}{60} \times 100\% \tag{2}$$

where $n$ is the electron transfer number, $F$ is Faraday constant (96,485 C mol$^{-1}$), $x$ is the mole fraction of the product, $\upsilon$ is the flow rate

of gas (sccm), $V_m$ is the molar volume (24.5 L mol$^{-1}$), and $I$ is the applied current (A).

For simulated crude ethylene, because of the abundant ethylene in the feed gas, the FE of $C_2H_4$ was calculated as follows:

$$FE_{ethylene}(\%) = (1 - FE_{otherproducts}) \times 100\% \tag{3}$$

where $FE_{other\ products}$ represent the total FE of other products exclude $C_2H_4$, including $H_2$, $C_2H_6$, and $C_4$ olefin.

The theoretical limiting conversion current ($I_{limit}$) for completely reduce $C_2H_2$ to $C_2H_4$ was calculated as follows:

$$I_{limit} = \frac{nF\upsilon x}{V_m \times 60} \tag{4}$$

where $n$ is the electron transfer number (2 for $C_2H_2$ reducing to $C_2H_4$), $F$ is Faraday constant (96,485 C mol$^{-1}$), $\upsilon$ is the flow rate of gas (sccm), $x$ is the mole fraction of the $C_2H_2$ and $V_m$ is the molar volume (24.5 L mol$^{-1}$). For example, in our $C_2H_2$ impurities removal experiments using a two-electrode MEA-type reactor (electrode surface area, 4 cm$^2$), $x = 0.5\%$ and $\upsilon = 10.64$ sccm (10 sccm in Ar mode), thus $I_{limit} = 6.98$ mA.

The acetylene conversion and selectivity for ethylene were calculated as follows:

$$Conversion(\%) = \frac{c - c'}{c} \times 100\% \tag{5}$$

$$Selectivity(\%) = \frac{c - c' - [C_2H_6] - [C_4]}{c - c'} \times 100\% \tag{6}$$

where $c$ represents the acetylene concentration in the feed gas and $c'$, $[C_2H_6]$, $[C_4]$ are the concentration of acetylene, ethane, and $C_4$ olefins in the product gas.

The liquid products were quantified by collecting and analyzing the electrolyte using the 400 MHz NMR spectrometer. Typically, 600 μl electrolyte was mixed with 100 μl $D_2O$ (Sigma Aldrich, 99.9 at.% D) and 0.05 μl dimethylsulfoxide (Sigma Aldrich, 99.9%) as internal standard.

## Characterizations

Scanning electron microscope (SEM) images were taken on Gemini SEM 300 (ZEISS, Germany) at an accelerating voltage of 5 kV. X-ray diffraction (XRD) patterns were recorded using the Shimadzu X-ray Diffractometer (XRD-6100, Japan) with Cu-K$_\alpha$ radiation ($\lambda = 1.54178$ Å). X-ray photoelectron spectroscopy (XPS) measurements were performed on a Kratos-Axis Supra XPS spectrometer with an exciting source of Al K$_\alpha$ = 1486.6 eV. The binding energies obtained in the XPS spectral analysis were corrected by referencing C 1$s$ to 284.3 eV. Transmission electron microscope (TEM), high-resolution transmission electron microscope (HRTEM), high-angle annular dark-field scanning transmission electron microscope (HAADF-STEM) images, and energy dispersive X-ray (EDX) elemental mapping were carried out on Tecnai G$^2$ F20 S-TWIN using Mo-based TEM grids. Fourier transform infrared spectroscopy (FTIR) spectra were obtained at room temperature on a Thermo Fisher Nicolet iS50 ATR spectrometer equipped with an MCT detector. The X-ray absorption spectra (XAS) of Cu K-edges were conducted at BL11B beamline of Shanghai Synchrotron Radiation Facility (SSRF) under "top-up" mode with a constant current of 200 mA, recorded under fluorescence mode with a Lytle detector.

## Computational details

The structural optimizations were performed with density functional theory, with a periodic plane-wave implementation using Vienna ab initio Simulation Package (VASP) code[40,41]. The exchange-correlation energy was modelled by using Perdew–Burke–Ernzerhof (PBE)

functional[42] within the generalized gradient approximation (GGA). The projector augmented wave (PAW) pseudo-potentials[43] were used to describe ionic cores. An energy cutoff of 500 eV was adopted. A first-order Methfessel–Paxton smearing of 0.1 eV was applied to the orbital occupation during the geometry optimization and for the energy computations.

The adsorption energies were evaluated using three-layer $3 \times 3$ supercells with the bottom two layers constrained, and $[4 \times 4 \times 1]$ Monkhorst–Pack $k$-point grids were used[44] with a convergence threshold of $10^{-5}$ eV for the iteration in the self-consistent field (SCF). All structures were optimized until force components were <0.02 eV/Å. The vibrational frequencies of free molecules and adsorbates were calculated by using the phonon modules in the VASP 5.3 code. A standard thermodynamic correction was applied to determine the free energy corrections, including the correction of the effect from zero-point energy, pressure, inner energy, and entropy.

The transition states were determined using the method of climbing image nudged elastic band (CI-NEB)[45], with a convergence threshold of 0.05 eV/Å. All structures contained a three-layer metal slab with atoms in the top layer relaxed and the rest fixed, along with an ice-like water structure[46] for the (111) facets and hydrogen-bonded water layers for the (211) facet determined through minima hopping[47].

The potential-dependent electrochemical kinetic barriers were obtained through the charge-extrapolation scheme[48,49]. All transition states were referenced to the initial state of aqueous protons and electrons in bulk solution, as determined using the computational hydrogen electrode[50].

## Data availability
All the data that support the findings of this study are available from the corresponding authors upon reasonable request. Source data are provided with this paper.

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

## Acknowledgements

J.Z. acknowledges the National Key Research and Development Program of China (2021YFA1500500, 2019YFA0405600), CAS Project for Young Scientists in Basic Research (YSBR-051), National Science Fund for Distinguished Young Scholars (21925204), NSFC (U19A2015, 22221003, 22250007), Fundamental Research Funds for the Central Universities, Provincial Key Research and Development Program of Anhui (202004a05020074), K.C. Wong Education (GJTD-2020-15), Collaborative Innovation Program of Hefei Science Center, CAS (2022HSC-CIP004), and the DNL Cooperation Fund, CAS (DNL202003). C.X. acknowledges the NSFC (22102018 and 52171201), the Natural Science Foundation of Sichuan Province (2022NSFSC0194), the "Pioneer" and "Leading Goose" R&D Program of Zhejiang (No. 2023C03017), the Hefei National Research Center for Physical Sciences at the Microscale (KF2021005), and the University of Electronic Science and Technology of China for startup funding (A1098531023601264). T.Z. acknowledges the NSFC (22005291 and 22278067), the Natural Science Foundation of Sichuan Province (2023NSFSC0094) and the University of Electronic Science and Technology of China for startup funding (A1098531023601356). X.L. acknowledges the NSFC (22109082). C.L. acknowledges the Sichuan Natural Science Foundation for Young Scholars (2023NSFSC0911). We thank beamline BL11B of Shanghai Synchrotron Radiation Facility for providing the beamtime. This work was partially carried out at the USTC Center for Micro and Nanoscale Research and Fabrication.

## Author contributions

The project was conceptualized and supervised by C.X., J.Z., and T.Z. W.X. prepared the catalysts, performed the catalytic tests and in situ electrochemical experiments with the help of C.L., X.Z., J.L., and Z.Y. W.X. performed the catalyst characterizations. X.L. and H.P. carried out the DFT calculations. W.X. and P.C. performed the XAFS measurements. P.X. designed the two-electrode MEA-type reactor. H.L. and Q.J. helped in the analysis of data. W.X., X.L., T.Z., C.X., and J.Z. wrote the manuscript with input from all authors. All authors discussed the results and commented on the manuscript.

## Competing interests

The authors declare no competing interests.
