## [Peer Review File · Nature Communications]

Electrosynthesis of polymer-grade ethylene via acetylene semihydrogenation over undercoordinated Cu nanodotsREVIEWER COMMENTS

Reviewer #1 (Remarks to the Author):

The paper reports on a scorching topic in literature, i.e. C₂H₂ semihydrogenation and several papers were published in the last year on this field of research. Some important findings were already achieved in the literature such as the adoption of a GDE electrode that is also considered in this paper. In this sense, this paper is of relevance and suitable for the journal, especially for the impressive FE obtained for the C₂H₂ semihydrogenation, but the adoption of GDE and Copper is something so far well established. Copper nanodots for the reduction (hydrogenation) of acetylene have been prepared by reduction Cu₂Cl(OH)₃ nanoparticle in presence of acetylene itself. The nature of nanodots was determined by combining HAADF-STEM, STEM-EDX, HRTEM, XPS, XRD and in-situ XAS revealing that metallic copper is present in unsaturated form (CN 10). Under pure acetylene flow in 1M KOH the FE was about 96% with the major by-product being a C₄. The unsaturated nature of those NDs is the key to improve selectivity, since the competing HER is suppressed, while on microparticles this does not append. This is confirmed also by DFT calculation coupled with in situ Raman and IR to elucidate the mechanism. Finally, simulated raw ethylene was used and purified in flow cell with a MEA. In optimized conditions, a selectivity of 93% was obtained.

Other Comments

Authors say that Cu NDs and Cu MPs are in the form of metallic particles, they compare the white line of Cu NDs at different applied potentials and they say that it resembles that of copper foil, suggesting that Cu in NDs catalyst maintained a metallic feature under EASH condition. However, it is my impression that the feature of Cu₂O can be observed. Furthermore, according to the Pourbaix diagram, being the pH alkaline, copper in the experimental pH/potential conditions is expected to be in the form of hydroxide, or at least the surface should present some oxide/hydroxide. If so a Raman response should be observed but nothing is declared by authors perhaps because of a low surface concentration. In any case, the metallic nature of the active site is far to be convincing and so I wonder whether some more accurate analysis of the in situ XANES spectra of the Cu NDs can be provided (no further measurements but deconvolution of those presented)

Are there any evidence of Copper leaching after the work loading?

Overall I suggest accepting the paper but some more convincing insights on active sites need to be provided

Reviewer #2 (Remarks to the Author):

Reviewer Report

TITLE: Electrosynthesis of polymer-grade ethylene using undercoordinated Cu nanodots

AUTHORS: Xue and co-workers

SUBMITTED TO: Nature Communications

This work investigates the application of Cu nanodots as catalysts for the electrochemical partial hydrogenation of acetylene. The experimental data is corroborated by modeling, addressing coordination-dependent relationships in the electrocatalytic reaction. This manuscript is interesting and relevant for the readership of Nature Communications. If the authors can address the following problems, I believe the manuscript should be publishable.

1. "Electrosynthesis of polymer-grade ethylene" in the title is kind of misleading, suggesting that the authors synthesizing C₂H₄. Instead, what is actually conducted is electrochemical partial hydrogenation of acetylene. It would be beneficial to rephrase the title.
2. How about the mass loading of Cu NDs on GDE? As far as I understood, it was fixed to be 1 mg/cm². Was this parameter (mass loading per geometric area) studied in more detail?
3. It would be beneficial if the authors would provide more physicochemical characteristics of the precursor Cu₂Cl(OH)₃ nanoparticles and their transformation into metallic Cu NDs.
4. The paper would be aided by more discussion about the preparation method to help the reader follow the text. Laterally, some discussion of the associated chemistry and scalability/cost of the method would be of benefit to emphasize widespread applicability.
5. Electron microscopy data should be supported by (selected area) electron diffraction.
6. In the present study, the existence of Cu sites with unsaturated coordination is central. Please discuss the formation mechanism of such sites and the synthesis parameters that govern such Cu appearance. Is unsaturated coordination result from thermodynamic or kinetic? Also, it would be interesting to know if there are any electronic effects in the Cu NDs that occur owing to the lower coordination number.
7. This may be the subject of future work but is there a correlation between Cu NDs' loading on GDE and selectivity towards ethylene or have you already reached the maximum? In other words, if you decreased the loading below 29%, could you improve selectivity further, for example, by providing a higher degree of Cu sites with unsaturated coordination?
8. Please provide a detailed characterization of the Cu NDs electrocatalyst after stability testing (e.g., Fig. 4c or Fig. 4e).
9. The main problem of the paper is that the discussion part is totally missing. For the obtained high-performance Cu ND electrocatalyst, it is recommended that the author add the discussion together with the comparison of its electrocatalytic performance with other reported electrochemical conversions of C₂H₂ to C₂H₄, and look forward to its scientific and industrial prospects.

Review Sent Date: 28-Nov-2022

Point-by-point response to reviewer comments

Manuscript ID: NCOMMS-22-44099

Title: “Electrosynthesis of polymer-grade ethylene via acetylene semihydrogenation over undercoordinated Cu nanodots”

We thank the reviewers for their constructive comments which have helped us to greatly improve our research and the quality of our manuscript. We have now performed substantial experiments and included additional analysis to fully address the reviewers’ concerns and suggestions. Below, we address the points raised by reviewers one by one.

Reviewer 1

The paper reports on a scorching topic in literature, i.e. C₂H₂ semihydrogenation and several papers were published in the last year on this field of research. Some important findings were already achieved in the literature such as the adoption of a GDE electrode that is also considered in this paper. In this sense, this paper is of relevance and suitable for the journal, especially for the impressive FE obtained for the C₂H₂ semihydrogenation, but the adoption of GDE and Copper is something so far well established. Copper nanodots for the reduction (hydrogenation) of acetylene have been prepared by reduction Cu₂Cl(OH)₃ nanoparticle in presence of acetylene itself. The nature of nanodots was determined by combining HAADF-STEM, STEM-EDX, HRTEM, XPS, XRD and in-situ XAS revealing that metallic copper is present in unsaturated form (CN 10). Under pure acetylene flow in 1M KOH the FE was about 96% with the major by-product being a C₄. The unsaturated nature of those NDs is the key to improve selectivity, since the competing HER is suppressed, while on microparticles this does not append. This is confirmed also by DFT calculation coupled with in situ Raman and IR to elucidate the mechanism. Finally, simulated raw ethylene was used and purified in flow cell with a MEA. In optimized conditions, a selectivity of 93% was obtained. Overall, I suggest accepting the paper but some more convincing insights on active sites need to be provided.

Response

We sincerely appreciate the reviewer’s thoughtful evaluation of our manuscript, the succinct summary of our key research results, the positive comments about our work,

and the insightful suggestions, which helped us improve the quality of our manuscript. To provide more convincing insights on active sites of Cu NDs over EASH, we have supplemented more experiments and discussions, which are presented in our point-by-point response as below.

Comment 1

Authors say that Cu NDs and Cu MPs are in the form of metallic particles, they compare the white line of Cu NDs at different applied potentials and they say that it resembles that of copper foil, suggesting that Cu in NDs catalyst maintained a metallic feature under EASH condition. However, it is my impression that the feature of Cu₂O can be observed. Furthermore, according to the Pourbaix diagram, being the pH alkaline, copper in the experimental pH/potential conditions is expected to be in the form of hydroxide, or at least the surface should present some oxide/hydroxide. If so a Raman response should be observed but nothing is declared by authors perhaps because of a low surface concentration. In any case, the metallic nature of the active site is far to be convincing and so I wonder whether some more accurate analysis of the in situ XANES spectra of the Cu NDs can be provided (no further measurements but deconvolution of those presented).

Response

We thank the reviewer for this very important suggestion.

Firstly, *in situ* Raman measurements in the range of 200 to 1000 cm⁻¹ using 532-nm laser were conducted to analyze the valence state evolution of copper during electrocatalytic C₂H₂ semihydrogenation. As shown in Fig. R1, however, no Raman fingerprint signals of Cu-O or Cu-OH was observed at open-circuit, nor at working potentials [*Angew. Chem. Int. Ed.* **61**, 4 (2022)], which was probably due to low surface concentration of Cu.

After that, we sought to deconvolve the representative *in situ* X-ray absorption near-edge structure (XANES) spectra in Fig. 1e by linear combination fitting of the spectra of Cu⁰ and Cu⁺ according to reviewer's suggestion to further resolve the active sites. As revealed in Fig. R2 and Table R1, the percentage of Cu⁰ in Cu NDs was 99.90% at the applied potential of -0.7 V vs. RHE (*under maximum C₂H₄ FE*). Taken above together with other characterization results mentioned in the manuscript (Fig. 1d-f, Supplementary Fig. 6, and Supplementary Fig. 8), we believe the active sites of Cu NDs

in electrocatalytic acetylene semihydrogenation were undercoordinated metallic Cu nanodots.

We have added these new results and discussion into the revised manuscript (line 2, page 6) and SI (page S10).

Fig. R1 | *In situ* electrochemical Raman spectra of Cu NDs in a 1 M KOH solution at different working potentials vs. RHE without IR compensation. OCP, open-circuit potential.

Fig. R2 (now added as Supplementary Fig. 9 in the revised SI) | Linear combination fitting for derivative of Cu K-edge XANES over Cu NDs at -0.7 V vs. RHE, corresponding to the data in Fig. 1e. Fitting range was between -20 eV to 40 eV, relative to E_0 . Black and red solid lines denote experimental and fitting data, respectively. Fractions of Cu foil and Cu_2O making up the fitted spectra are shown by short dashed line.

Table R1 | Linear combination fitting results corresponding to Fig. R2.

Sample	Weight of Cu^0 (%)	Weight of Cu^+ (%)	Chi-square	R-factor
Cu NDs -0.7 V	99.90	0.10	0.0022	0.0321

Comment 2

Are there any evidence of Copper leaching after the work loading?

Response

We appreciate the reviewer for this valuable comment.

We have evaluated copper leaching of Cu NDs in a typical flow cell system by conducting inductively coupled plasma atomic emission spectroscopy (ICP-AES) analysis towards post-reaction electrolyte. 20 mL of 1 M KOH electrolyte was flowed through cathode and circulated by a peristaltic pump at a flow rate of 4 mL min^{-1} . In the process of constant current electrolysis at -50 mA cm^{-2} (Fig. R3a), 2 ml of electrolyte was taken out periodically and 1 M HCl was added for acidification. As depicted by Fig. R3b, only $5.28 \text{ }\mu\text{g}$ of Cu was leached (equating to a 2.35% loss of total Cu at the working electrode) after 16 hours, suggesting our Cu NDs exhibited excellent stability under EASH electrolysis.

We have added the results into the revised manuscript (line 1, page 7) and SI (page S17).

Fig. R3 (now added as Supplementary Fig. 16 in the revised SI) | Study on copper leaching under working condition in a flow cell. (a) Cathode potential against time at current density of -50 mA cm^{-2} over Cu NDs. **(b)** Mass of leached Cu at different time.

Reviewer 2

This work investigates the application of Cu nanodots as catalysts for the electrochemical partial hydrogenation of acetylene. The experimental data is corroborated by modeling, addressing coordination-dependent relationships in the electrocatalytic reaction. This manuscript is interesting and relevant for the readership of Nature Communications. If the authors can address the following problems, I believe the manuscript should be publishable.

Response

We highly appreciate the reviewer for time and insightful comments on our work.

Comment 1

“Electrosynthesis of polymer-grade ethylene” in the title is kind of misleading, suggesting that the authors synthesizing C₂H₄. Instead, what is actually conducted is electrochemical partial hydrogenation of acetylene. It would be beneficial to rephrase the title.

Response

We thank the reviewer for this important suggestion. We have modified the title from “Electrosynthesis of polymer-grade ethylene using undercoordinated Cu nanodots” to “Electrosynthesis of polymer-grade ethylene via acetylene semihydrogenation over undercoordinated Cu nanodots”.

Comment 2

How about the mass loading of Cu NDs on GDE? As far as I understood, it was fixed to be 1 mg/cm². Was this parameter (mass loading per geometric area) studied in more detail?

Response

We appreciate the reviewer for raising this significant comment.

The theoretical mass loading of catalysts precursor on the gas diffusion electrode was 1 mg cm⁻² in all our experiments, and due to the mass loss during the spraying of

inks, the finally obtained mass loading of catalyst precursor used for the test in Fig. 2 was 0.89 mg cm^{-2} .

We have evaluated the effect of mass loading per geometric area on acetylene-to-ethylene performance by controlling the mass loading of precursor on GDE at 0.49, 0.89, and 2.02 mg cm^{-2} , respectively. After *in situ* electroreduction, the corresponding copper mass loading per geometric area were calculated to be 0.13, 0.25, and 0.56 mg cm^{-2} , respectively, using ICP-AES for reviewer's reference. As shown in Fig. R4, the faradaic efficiency (FE) of coupling product C_4 would be improved with the increase of Cu NDs mass loading on GDE. We assume the high mass loading results in impeded mass transfer, which promotes the surface coverage of adsorbed acetylene and thus the coupling to C_4 .

Fig. R4 (now added as Supplementary Fig. 12) | FEs of electrocatalytic acetylene semihydrogenation products at different current densities over Cu NDs with different catalysts mass loading on gas diffusion electrode. Three bars from left to right correspond to the mass loading of 0.49, 0.89, and 2.02 mg cm^{-2} for catalyst precursors, respectively. The error bars correspond to the standard deviation of at least three independent measurements.

Nevertheless, due to the fewer active sites, Cu NDs with the mass loading per geometric area of 0.49 mg cm^{-2} need a more negative potential to achieve the same ethylene partial current density as the Cu NDs with a mass loading of 0.89 mg cm^{-2} (Fig. R5a). To rule out the mass loading effect and get a clear idea of the intrinsic activity, we further calculated the mass activity by normalizing the ethylene partial current density to the mass loading of Cu NDs. As shown in Fig. R5b, apparently, Cu NDs with the mass loading of 0.89 mg cm^{-2} for precursor possessed the highest mass activity under the same potential.

Fig. R5 (now added as Supplementary Fig. 13) | Variation in the (a) ethylene partial current density and (b) mass normalized ethylene partial current density against applied potential over Cu NDs with different catalysts mass loading on gas diffusion electrode. The error bars correspond to the standard deviation of at least three independent measurements.

Thus, we controlled the mass loading of catalysts precursors on the gas diffusion electrode to be ~ 1 mg cm⁻².

We have added these new results and discussion into the revised manuscript (line 10, page 14) and SI (page S13, S14).

Comment 3

It would be beneficial if the authors would provide more physicochemical characteristics of the precursor $Cu_2Cl(OH)_3$ nanoparticles and their transformation into metallic Cu NDs.

Response

We thank the reviewer for this valuable comment.

We have provided HRTEM image of $Cu_2Cl(OH)_3$ precursor in Fig. R6 (now added as Supplementary Fig. 3). The lattice fringe spacing of 0.548 nm and 0.279 nm marked on the picture can be indexed to the (101) and (113) plane of $Cu_2Cl(OH)_3$, respectively. Beyond that, we have conducted XRD, SEM, TEM, XPS and FTIR analysis towards $Cu_2Cl(OH)_3$ precursor in the initial manuscript (Supplementary Fig. 1, 2, 4, and 5).

In order to study the transformation of $Cu_2Cl(OH)_3$ into metallic Cu NDs under operation conditions, we carried out *in situ* Raman measurements in the range of 200 to 1000 cm⁻¹. However, because of the low surface concentration of copper, no Raman characteristic signal of Cu-O, Cu-OH or any change in the spectra was observed (Fig. R7). Nevertheless, we have already confirmed the transformation of $Cu_2Cl(OH)_3$ into

metallic Cu NDs by *in situ* XANES (Fig. 1e), XPS spectra (Supplementary Fig. 4 and 8), and XRD patterns (Supplementary Fig. 1 and 6) in the original manuscript, which were copied here for the reviewer's reference (Fig. R8 to R10).

Fig. R6 (now added as Supplementary Fig. 3) | HRTEM image of the precursor $\text{Cu}_2\text{Cl}(\text{OH})_3$ nanoparticles.

Fig. R7 | *In situ* electrochemical Raman spectra of Cu NDs in a 1 M KOH solution at different working potentials vs. RHE without IR compensation. OCP, open-circuit potential.

Fig. R8 | Normalized *in situ* XANES spectra of the Cu NDs catalyst under EASH conditions at different working potentials vs. RHE, along with the spectra of Cu foil, Cu₂O and CuO as references. OCP, open-circuit potential.

Fig. R9 | XPS spectra of the precursor and Cu NDs. (a) Cu 2p spectrum of precursor Cu₂Cl(OH)₃. (b) Cu 2p spectrum of Cu NDs. (c) Cu LMM auger spectrum of the Cu NDs.

Fig. R10 | XRD patterns of the precursor and Cu NDs. (a) XRD pattern of precursor Cu₂Cl(OH)₃ with 2theta from 20° to 80°. (b) Enlarged region between 40° and 80° of the XRD pattern of Cu NDs.

Comment 4

The paper would be aided by more discussion about the preparation method to help the reader follow the text. Laterally, some discussion of the associated chemistry and scalability/cost of the method would be of benefit to emphasize widespread applicability.

Response

We appreciate the reviewer for this valuable suggestion.

We have added more details and discussion about the preparation method into the revised manuscript (line 16, page 13) and copied here for the reviewer's reference.

“For the synthesis of carbon supported $\text{Cu}_2\text{Cl}(\text{OH})_3$ precursor, 400 mg of activated carbon (XC-72) was dispersed in 80 ml of ethanol in a 250 ml flask with sonication for 30 min. After that, 1 g of CuCl_2 was added into the flask and stirred for another 30 min to get a uniform blackish green ink. DEA solution (1 g of DEA dissolved in 5 ml of ethanol) was then slowly dropped into the flask in 10 min and refluxed at 100 °C for 1 h, under stirring. After cooling to room temperature, 100 ml of DI water was added into the flask and sonicated for 30 min to completely hydrolyze copper complex. The precursor was collected by centrifugation at 12,000 rpm for 10 min and washed three times with DI water and two times with ethanol, respectively, and then dried at 70 °C for 24 h. Cu NDs catalyst was then in situ formed by reducing the precursors at a constant current density of -100 mA cm^{-2} for 30 min in a standard three-electrode flow cell system supplied with C_2H_2 gas at a flow rate of 30 sccm, using 1 M KOH as electrolyte. The mass loading of copper in the Cu NDs can be controlled easily by adjusting the amount of CuCl_2 in the feed.”

Table R2 | Price of chemicals and the power of equipment in the experiment.

Chemicals	Price	Equipment	Power
activated carbon	0.27 USD g^{-1}	sonicator	200 W
Ethanol (>99.7%)	8.94 USD L^{-1}	stirring heater	825 W
Diethanolamine (>99%)	1.13 USD kg^{-1}	centrifuge	1,010 W
copper chloride(>99%)	50.60 USD kg^{-1}	drying oven	1,550 W

Overall, Cu NDs was formed by *in situ* electroreduction of carbon supported $\text{Cu}_2\text{Cl}(\text{OH})_3$ which achieved from the hydrolysis of CuCl_2 in the presence of DEA. According to the preparation method mentioned above, we could obtain about 450 mg catalyst for one time. Thus, we could easily get sufficient Cu NDs by adding more batches or scaling up the reactor. Furthermore, we have analyzed the cost of

synthesizing Cu NDs in the laboratory on the basis of the recipe above. The price of the chemicals (data from MACKLIN) and the power of equipment were listed in Table R2 for calculation.

The chemical cost is:

$$\begin{aligned} \text{Chemical cost} &= \text{activated carbon} + \text{diethanolamine} + \text{copper chloride} + \text{ethanol} \\ &= \frac{\$0.27}{g} * \frac{400\text{mg} * 1g}{1,000\text{mg}} + \frac{\$1.13}{kg} * \frac{1g * 1kg}{1,000g} + \frac{\$50.60}{kg} * \frac{1g * 1kg}{1,000g} + \frac{\$8.94}{L} \\ &\quad * \frac{(85 + 240)\text{mL} * 1L}{1,000\text{mL}} = \$3.07 \end{aligned}$$

The electricity cost of the equipment is calculated from the power and the price of electricity, which is supposed to be 0.03 USD per kilowatt-hour.

Equipment electricity cost

$$\begin{aligned} &= \text{sonicator} + \text{stirring heater} + \text{centrifuge} + \text{drying oven} \\ &= \frac{\$0.03}{kWh} * (200W * 1h + 825W * 1.5h + 1,010W * 0.8h + 1,550W * 24h) \\ &= \$1.18 \end{aligned}$$

Experimentally, the full cell potential and current density were 2 V and 100 mA cm⁻², respectively. Assuming the synthesized 450 mg of precursor was loaded on the gas diffusion electrode with a spray loss of 1/2 and a mass loading of 1 mg cm⁻², the electricity cost of *in situ* electrolysis is:

$$\begin{aligned} \text{Electrolysis electricity cost} &= 450\text{mg} * \frac{1}{2} * \frac{1\text{cm}^2}{\text{mg}} * \frac{100\text{mA}}{\text{cm}^2} * 2V * 0.5h * \frac{\$0.03}{kwh} \\ &= \$0.0007 \end{aligned}$$

Thus, the total cost is:

$$\begin{aligned} \text{Total cost} &= \text{chemical cost} + \text{electricity cost} = \$3.07 + \$1.18 + \$0.0007 \\ &= \$4.25 \end{aligned}$$

Take above together, the cost of Cu NDs loaded gas diffusion electrode with an area of 225 cm² and a mass loading of 1 mg cm⁻² is 4.25 USD (equivalent to 189 USD m⁻²), which will be further decreased when the synthesis is scaled up. Thus, our preparation method is facile, cost-effective, and easy to scale up.

Comment 5

Electron microscopy data should be supported by (selected area) electron diffraction.

Response

We thank the reviewer for this thoughtful suggestion.

We have provided the selected area electron diffraction pattern of Cu NDs in Fig. R11 (now added as Supplementary Fig. 7). The pattern showed the presence of diffused rings, indicating the polycrystalline nature of the Cu NDs. As specified in the diffraction pattern, the bright fringes were assigned to the (220), (200), (111) plane of copper, and (002) plane of activated carbon, respectively.

Fig. R11 (now added as Supplementary Fig. 7) | Selected area electron diffraction pattern of Cu NDs.

Comment 6

In the present study, the existence of Cu sites with unsaturated coordination is central. Please discuss the formation mechanism of such sites and the synthesis parameters that govern such Cu appearance. Is unsaturated coordination result from thermodynamic or kinetic? Also, it would be interesting to know if there are any electronic effects in the Cu NDs that occur owing to the lower coordination number.

Response

We appreciate the reviewer for this insightful comment.

The formation of unsaturated metal sites can happen in different stages of the reaction. In the $\text{Cu}_2\text{Cl}(\text{OH})_3$ precursor formation stage, the coordination complex is typically formed from the interaction of metal ions with ligands, which consist of multiple donor atoms. These complexes can have unsaturated metal sites if there are fewer ligands bound to the metal ion than the maximum number it can coordinate with. Such a phenomenon has been widely reported for the fine-tuning of unsaturated metal

sites of metal–organic frameworks [*Nature Chemistry* **11**, 622-628 (2019)] and is thermodynamically controlled. On the other side, as a result of the high surface energy, unsaturated metal sites are easy to migrate and aggregate in the process of synthesis to form flat or non-defective surface [*Chem. Rev.* **120**, 11900-11955 (2020)]. Typically, we can use appropriate supports to prevent migration and agglomeration of the unsaturated sites [*Nat. Commun.* **12**, 2932 (2021)]. In the preparation of Cu NDs, activated carbon was used to isolate and distribute Cu sites, which is also known as a thermodynamic process. Then, during the electrochemical reduction process, the number of electrons added to the metal ion during reduction is insufficient to fully satisfy its coordination number, resulting in the formation of metal atoms with unsaturated coordination sites with a partially filled valence shell. This can also occur when the precursor has full coordination numbers. The number of electrons added to the metal ion during reduction is possibly insufficient to fully satisfy its coordination number, resulting in the formation of a metal atom with unsaturated coordination sites. This process is more likely to be kinetically controlled. Overall, the formation of such Cu nanodots with a feature of undercoordinated sites is a synergistic effect of thermodynamics and kinetics.

As unsaturated metal sites play a key role in many important processes, including catalysis, biochemistry, and environmental remediation, this is an important area of research in the fields of chemistry and materials science. The formation mechanism of unsaturated metal sites involves various physical and chemical interactions between the metal atoms and their surrounding environment. This process is influenced by factors such as temperature, ligand concentration, metal ion concentration, reduction potentials, PH, and the presence of other elements in the system. In our system, as reported previously [*Nat. Commun.* **12**, 2932 (2021)], we can also adjust the Cu-Cu bond coordination number easily by controlling the proportion of activated carbon in the feed.

Because of the orbital overlapping between metal atoms, the electronic structure of metal with unsaturated coordination is more complicated [*Chem. Rev.* **118**, 4981-5079 (2018)]. In our system, due to the special electronic effect, Cu sites with unsaturated coordination exhibited significantly weaker stabilization to $C_2H_2^*$ than non-defective surface and thus it was easy to reduce adsorbed $C_2H_2^*$ to $C_2H_3^*$ over Cu NDs which was consistent with the in situ differential electrochemical mass spectrometry results (Fig. 2d). What is more, the kinetic barrier of proton transfer to $C_2H_2^*$ and forming $C_2H_3^*$ was lower than that of transferring a proton to form H^* and

producing hydrogen and thus Cu NDs possessed higher electrocatalytic acetylene semihydrogenation activity than HER. According to experimental and theoretical results mentioned above, Cu NDs with undercoordinated sites own special electronic effect compared with non-defective Cu surface.

Comment 7

This may be the subject of future work but is there a correlation between Cu NDs' loading on GDE and selectivity towards ethylene or have you already reached the maximum? In other words, if you decreased the loading below 29%, could you improve selectivity further, for example, by providing a higher degree of Cu sites with unsaturated coordination?

Response

We thank the reviewer for this visionary comment.

Fig. R12 (now added as Supplementary Fig. 17) | FEs of electrocatalytic acetylene semihydrogenation products at different current densities over Cu NDs with different copper mass loading. Cu NDs catalysts with copper mass loading of 29.04 wt%, 18.30 wt% and 8.74 wt% were synthesized by changing CuCl₂ feeding amount while maintaining the quantity of activated carbon the same. Three bars from left to right correspond to the copper mass loading of 29.04 wt%, 18.30 wt% and 8.74 wt%, respectively. The error bars correspond to the standard deviation of at least three independent measurements.

We have synthesized Cu NDs catalysts with copper mass loading of 18.30 wt% and 8.74 wt% by reducing CuCl₂ feeding amount while maintaining the quantity of activated carbon the same. As shown in Fig. R12, the selectivity of C₂H₄ would improve

with the decrease of copper mass loading which might provide a higher degree of Cu sites with unsaturated coordination [*Nat. Commun.* **12**, 2932 (2021)]. However, decrease of the copper active sites density on the surface would weaken the acetylene semihydrogenation catalytic activity as well. Instead, more hydrogen evolution reaction happened over the catalysts with a lower copper mass loading. For the reference of reviewer, we listed the selectivity of C₂H₄ and C₄ along with the FE of H₂ in Table R3.

We have added these results and discussions into the revised manuscript (line 2, page 7) and SI (page S18).

Table R3 | Selectivities of electrocatalytic acetylene semihydrogenation products at different current densities over Cu NDs with different copper mass loading.

Current density	Sample	Selectivity of C ₂ H ₄	Selectivity of C ₄	FE of H ₂
-100 mA cm ⁻²	Cu NDs-29.04 wt%	89.92%	10.07%	0.00%
	Cu NDs-18.30 wt%	91.22%	8.78%	0.01%
	Cu NDs-8.74 wt%	93.19%	6.76%	3.94%
-200 mA cm ⁻²	Cu NDs-29.04 wt%	90.81%	9.19%	0.00%
	Cu NDs-18.30 wt%	94.13%	5.87%	1.11%
	Cu NDs-8.74 wt%	96.26%	3.60%	7.70%
-300 mA cm ⁻²	Cu NDs-29.04 wt%	92.29%	7.71%	0.00%
	Cu NDs-18.30 wt%	95.74%	4.06%	4.00%
	Cu NDs-8.74 wt%	97.32%	2.61%	13.02%
-400 mA cm ⁻²	Cu NDs-29.04 wt%	93.88%	6.09%	0.11%
	Cu NDs-18.30 wt%	97.79%	2.14%	6.50%
	Cu NDs-8.74 wt%	98.81%	1.11%	18.64%
-500 mA cm ⁻²	Cu NDs-29.04 wt%	93.22%	6.03%	3.65%
	Cu NDs-18.30 wt%	98.48%	1.47%	12.94%
	Cu NDs-8.74 wt%	99.08%	0.83%	25.27%

Comment 8

Please provide a detailed characterization of the Cu NDs electrocatalyst after stability testing (e.g., Fig. 4c or Fig. 4e).

Response

We appreciate the reviewer for this valuable comment. We have repeated the stability test in Fig. 4c and characterized the Cu NDs electrocatalyst after stability test

in detail. As shown in Fig. R13, Cu NDs maintained its morphology, phase, and valence state after the stability test, indicating the excellent material stability of Cu NDs.

We have added these results and discussion into the revised manuscript (line 4, page 11) and SI (page S30).

Fig. R13 (now added as Supplementary Fig. 29) | Characterization of Cu NDs after the stability test. (a) TEM image. (b) XRD pattern of Cu NDs with 2theta from 40° to 80°. (c) Cu 2p XPS spectrum and (d) Cu LMM auger spectrum of Cu NDs.

Comment 9

The main problem of the paper is that the discussion part is totally missing. For the obtained high-performance Cu ND electrocatalyst, it is recommended that the author add the discussion together with the comparison of its electrocatalytic performance with other reported electrochemical conversions of C₂H₂ to C₂H₄, and look forward to its scientific and industrial prospects.

Response

We thank the reviewer for pointing out this issue.

We have already compared the intrinsic activity under pure acetylene flow without ethylene in Supplementary Fig. 22 and Supplementary table 3 (copied below). Cu NDs achieved an impressive C_2H_4 partial current density exceed 450 mA cm^{-2} with C_2H_4 FE of 90.4%, outperforming the reported state-of-art C_2H_2 -to- C_2H_4 electrocatalysts, which indicated the outstanding intrinsic activity and selectivity towards electrocatalytic acetylene semihydrogenation over undercoordinated Cu sites.

Supplementary Fig. 22 | Comparison in the $FE_{C_2H_4}$ and ethylene partial current density over different reported catalysts. The translucent ball represents the catalyst performed in H-Cell.

Supplementary table 3 | Comparison of EASH performance under pure acetylene flow without ethylene.

Catalyst	FE (%)	$j_{C_2H_4}$ (mA cm^{-2})	Reactor	Reference
Cu NDs	95.94	-335.8	Flow Cell	This work
Cu NDs	90.42	-452.1	Flow Cell	This work
Cu dendrites	96.00	-134.0	Flow Cell	Ref. 1
LD-Cu	79.50	-41.4	Flow Cell	Ref. 2
NHC-Cu	98.00	-159.0	Flow Cell	Ref. 3
SA-Ni-NC	91.30	-84.2	Flow Cell	Ref. 4
CoPc	88.00	-150.8	Flow Cell	Ref. 5
Cu MPs	83.20	-24.1	H-Cell	Ref. 6
Cu	28.90	-4.6	H-cell	Ref. 7
Pt	22.00	-4.2	H-Cell	Ref. 8

[Ref. 1: *Nat. Catal.* **4**, 565-574 (2021); Ref. 2: *Nat. Catal.* **4**, 557-564 (2021); Ref. 3: *Nat. Commun.* **12**, 6574 (2021); Ref. 4: *J. Mater. Chem. A* **10**, 6122 (2022); Ref. 5: *Chem. Eng. J.* **431**, 134129 (2022); Ref. 6: *Nat. Commun.* **12**, 7072 (2021); Ref. 7: *Electrochem. Commun.* **34**, 90-93 (2013); Ref. 8: *J. Electrochem. Soc.* **118**, 236 (1971)]

To evaluate the acetylene impurity removal performance over Cu NDs, we also compared our work with reported electrocatalysts under simulated crude ethylene (Table R4). Superior to state-of-art reported electrocatalysts, Cu NDs could convert C₂H₂ impurity completely at a relatively high space velocity of 1.35×10⁵ ml g⁻¹ h⁻¹. In addition, Cu NDs was capable to operate over 130 h with a C₂H₄ selectivity of 94%.

In conclusion, Cu NDs not only possesses excellent intrinsic C₂H₂-to-C₂H₄ catalytic activity and selectivity, but also has ability to completely remove the acetylene impurity efficiently, indicating its scientific significance. We have added these results and discussion into the revised SI (page S23, page S39).

Table R4 (now added as Supplementary table 6) | Comparison of acetylene impurity removal performance between our work and reported electrocatalysts.

Catalyst	Space velocity (ml g ⁻¹ h ⁻¹)	Residual C ₂ H ₂ (ppm)	C ₂ H ₄ selectivity (%)	Stability (h)	Reference
Cu NDs	1.35×10 ⁵	0	94	130	This work
Cu dendrites	9.60×10 ⁴	4	97	120	Nat. Catal. 4 , 565-574 (2021)
LD-Cu	/	5	90.1	4	Nat. Catal. 4 , 557-564 (2021)
NHC-Cu	9.6×10 ⁵	30	99	100	Nat. Commun. 12 , 6574 (2021)
SA-Ni-NC	2.4×10 ⁴	264	99	8	J. Mater. Chem. A 10 , 6122 (2022)

In order to envision the industrial prospect of Cu NDs, we did a quantitatively techno-economic analysis of our strategy for crude ethylene purification according to previous method [*Ind. Eng. Chem. Res.* **57**, 2165-2177 (2018)], assuming a C₂H₄ purification rate of 100,000 kg per day.

Techno-economic analysis

According to Fig. 4e, electrode area of 25 cm² was required to remove acetylene impurity at a flow rate of 50 ml min⁻¹ and the current was 50 mA with a cell voltage of -1.89 V.

The total flow rate needed is:

$$\begin{aligned} \text{total flow rate} &= 100,000 \frac{\text{kg}}{\text{day}} * \frac{\text{day}}{1,440\text{min}} * \frac{\text{mol}}{28\text{g}} * \frac{1,000\text{g}}{\text{kg}} * \frac{24,500\text{ml}}{\text{mol}} \\ &= 60,763,889 \frac{\text{ml}}{\text{min}} \end{aligned}$$

The total electrode area needed is given by the required flow rate:

$$\text{total electrode area} = 60,763,889 \frac{\text{mL}}{\text{min}} * \frac{\text{min}}{50\text{ml}} * 25\text{cm}^2 * \frac{\text{m}^2}{10,000\text{cm}^2} = 3,038.2\text{m}^2$$

The total current needed for our system is:

$$total\ current = \frac{3,038.2m^2}{25cm^2} * \frac{10,000cm^2}{m^2} * 50mA * \frac{A}{1,000mA} = 6,0764A$$

The power needed is given from $P=UI$:

$$power = 1.89V * 60,764A * \frac{kW}{1,000W} = 114.8kW$$

For the full cell reaction of $2C_2H_2 + 2H_2O \rightarrow 2C_2H_4 + O_2$ or $2H_2O \rightarrow 2H_2 + O_2$ the water consumption rate is:

$$\begin{aligned} water\ consumption\ rate \\ &= 60,764A * \frac{2}{4e^- * 96,485 \frac{C}{mol}} * \frac{0.018kg}{mol} * \frac{0.2642gal}{kg} * \frac{86,400s}{day} \\ &= \frac{129.4gal}{day} \end{aligned}$$

Capital costs

From the DOE (U. S. Department of Energy) H₂A analysis (the hydrogen analysis project of DOE) for central grid electrolysis, the electrolyzer cost for the stack component is 250.25 USD per kilowatt. The reference electrolyzer is operated at 0.175 A cm⁻² and 1.75 V. The installation factor is 1.2. Thus, the cost per area for the reference electrolyzer is:

$$Ref.\ electrolyzer\ cost = \frac{\$250.25}{kW} * \frac{0.175A}{cm^2} * 1.75V * \frac{10^4cm^2}{m^2} * \frac{kW}{1,000W} * 1.2 = \frac{\$919.7}{m^2}$$

Thus, the electrolyzer capital cost is given by multiplying the total area:

$$Electrolyzer\ capital\ cost = 3,038.2m^2 * \frac{\$919.7}{m^2} = \$2,794,232.5$$

From the H₂A, the balance of plant (BoP) capital cost is 35% of the total cost, while the stack is 65%:

$$BoP\ capital\ cost = \$2,794,232.5 * \frac{0.35}{0.65} = \$1,504,586.7$$

Because of the hydrogen byproduct, the PSA (Pressure Swing Adsorption) capital cost is calculated by scaling the reference cost to the total flow rate:

$$PSA\ capital\ cost = \$1,989,043 * \left(\frac{60,763,889 \frac{ml}{min} * \frac{m^3}{1,000,000ml}}{1,000 \frac{m^3}{hr} * \frac{hr}{60min}} \right)^{0.7} = \$4,919,284.5$$

So, the sum of capital costs is \$9,218,103.7.

Operating costs

The electricity cost is calculated from the power and the price of electricity, which is supposed to be 0.03 USD per kilowatt-hour. Thus, the electricity cost for 1 year (350 days) is:

$$\text{Electricity cost} = 114.8\text{kW} * \frac{\$0.03}{\text{kWh}} * \frac{24\text{hr}}{\text{day}} * \frac{350\text{day}}{\text{year}} = \frac{\$28,929.6}{\text{year}}$$

The cost of the PSA is calculated by scaling the reference cost to the flow rate:

$$\text{PSA operating cost} = \frac{0.25\text{kWh}}{\text{m}^3} * 3,645.8 \frac{\text{m}^3}{\text{hr}} * \frac{8,400\text{hr}}{\text{year}} * \frac{\$0.03}{\text{kWh}} = \frac{\$229,685.4}{\text{year}}$$

The maintenance cost is assumed 2.5% of capital cost per year (from H₂A):

$$\begin{aligned} \text{Maintenance cost} &= (\$2,794,232.5 + \$1,504,586.7 + \$4,919,284.5) * \frac{0.025}{\text{year}} \\ &= \frac{\$230,452.6}{\text{year}} \end{aligned}$$

The cost of the water for 1 year is:

$$\text{water cost} = \frac{129.4\text{gal}}{\text{day}} * \frac{\$0.0054}{\text{gal}} * \frac{350\text{day}}{\text{year}} = \frac{\$244.6}{\text{year}}$$

The cost of the catalysts is included in the electrolyzer cost, thus, the total cost of C₂H₄ purification in the first 5 years is:

Total cost

$$\begin{aligned} &= \text{Capital cost} + \text{Operating cost} \\ &= \text{Capital cost} + (\text{electricity} + \text{PSA} + \text{maintenance} + \text{water}) * \text{year} \\ &= \$9,218,103.7 + (\$28,929.6 + \$229,685.4 + \$230,452.6 + \$244.6) * 5 \\ &= \$11,664,664.7 \end{aligned}$$

Take above together, in the case of the C₂H₄ purification rate of 100,000 kg per day, the total cost in the first 5 years is 11,664,664.7 USD, corresponding to 66.7 USD per ton. As an indispensable and essential part of ethylene production, the cost of ethylene purification in our system is just about 5.4% of ethylene price (\$1,230), indicating our system with Cu NDs as catalyst has the potential to industrialize. We have added the techno-economic analysis into the revised SI (Supplementary Note 1, page S40).

REVIEWERS' COMMENTS

Reviewer #1 (Remarks to the Author):

The authors have comprehensively responded to Reviewer observations, therefore I've no further questions and I suggest to accept the paper in the revised form

Reviewer #2 (Remarks to the Author):

I am quite happy with the answers of the authors, and hence, recommend the MS be published in the revised form.

Point-by-point response to reviewer comments

Manuscript ID: NCOMMS-22-44099A

Title: “Electrosynthesis of polymer-grade ethylene via acetylene semihydrogenation over undercoordinated Cu nanodots”

Reviewer #1:

“The authors have comprehensively responded to Reviewer observations, therefore I’ve no further questions and I suggest to accept the paper in the revised form.”

Response: We sincerely thank the reviewer.

Reviewer #2:

“I am quite happy with the answers of the authors, and hence, recommend the MS be published in the revised form.”

Response: We sincerely thank the reviewer.